# Are Supplements Consumed by Middle-Distance Runners Evidence-Based? A Comparative Study between Level of Competition and Sex

**DOI:** 10.3390/nu15224839

**Published:** 2023-11-20

**Authors:** Asier Del Arco, Aitor Martinez Aguirre-Betolaza, Ewa Malchrowicz-Mośko, Anna Gogojewicz, Arkaitz Castañeda-Babarro

**Affiliations:** 1Health, Physical Activity and Sports Science Laboratory, Department of Physical Activity and Sports, Faculty of Education and Sport, University of Deusto, 48007 Bilbao, Spain; asier.delarco@opendeusto.es (A.D.A.); a.martinezdeaguirre@deusto.es (A.M.A.-B.); 2Institute of Sport Sciences, Poznan University of Physical Education, 61-871 Poznań, Poland; malchrowicz@awf.poznan.pl; 3Institute of Health Sciences, Poznan University of Physical Education, 61-871 Poznań, Poland; gogojewicz@awf.poznan.pl

**Keywords:** middle-distance, supplementation, nutrition, performance, health

## Abstract

Background: Middle-distance running events have special physiological requirements from a training and competition point of view. Therefore, many athletes choose to take sport supplements (SS) for different reasons. To date, few studies have been carried out that review supplementation patterns in middle-distance running. The aim of the present study is to analyze the consumption of SS in these runners with respect to their level of competition, sex and level of scientific evidence. Methods: In this descriptive cross-sectional study, data was collected from 106 middle-distance runners using a validated questionnaire. Results: Of the total sample, 85.85% responded that they consumed SS; no statistical difference was found regarding the level of competition or sex of the athletes. With respect to the level of competition, differences were observed in the total consumption of SS (*p* = 0.012), as well as in that of medical supplements (*p* = 0.005). Differences were observed between sexes in the consumption of medical supplements (*p* = 0.002) and group C supplements (*p* = 0.029). Conclusions: Higher-level athletes consume SS that have greater scientific evidence. On the other hand, although the most commonly consumed SS have evidence for the performance or health of middle-distance runners, runners should improve both their sources of information and their places of purchase.

## 1. Introduction

Middle-distance running events are highly complex from a bioenergetic, training and tactical point of view [1]. The level of energy intensity is in a middle ground between aerobic and anaerobic metabolism [2], with the aerobic contribution in the 800 m being between 60 and 75% and slightly higher (77–85%) in the 1500 m [3]. In addition, due to the type of muscle fibers these athletes have (Mainly IIX and IIA [4]), most middle-distance runners can reach lactate peaks of >20 mmol/L, leading to muscle pH levels as low as 6.6 [5]. However, the high speed requirements make both aerobic and anaerobic metabolism contribute significantly during these events [6]. This can be reflected in the distribution of training intensities throughout the season. Middle-distance runners work a very wide spectrum of training zones, ranging from low-intensity running sessions to very-high-intensity glycolytic workouts [7]. In this way, elite middle-distance runners develop aerobic capacities similar to those of long-distance runners, mechanical skills close to those of sprinters, as well as a highly enhanced anaerobic capacity [1]. Some of these characteristics make them adopt different race strategies [8,9]. However, sometimes the difference between being a medalist or not is minimal [10], and the improvements seen with some SS are very worthwhile in terms of performance [11].

Supplements are defined as “A food, food component, nutrient, or nonfood compound that is purposefully ingested in addition to the habitually-consumed diet with the aim of achieving a specific health and/or performance benefit” [11]. Although many athletes use SS to improve their performance, there are other underlying reasons for their use [12]. According to the Australian Institute of Sport (AIS), supplements are classified into four groups using the “ABCD” system [13]. This is based on the latest scientific evidence for determining whether a product is safe, permitted and effective in improving performance or health: (A) supplements with solid scientific evidence in specific situations under established protocols; (B) components with emerging evidence that should be used in research or clinical settings; (C) supplements with limited evidence and effects on performance; (D) prohibited products or those with a high risk of contamination by doping substances. Regarding middle-distance races, some of the supplements that have shown the most evidence in improving performance are caffeine [14,15], β-Alanine [16,17,18] and sodium bicarbonate [19,20,21]. However, these SS are not among the most consumed by middle-distance runners, with the consumption of vitamins, minerals and amino acids being higher than the previously mentioned ones [22].

Although SS can provide both health and performance benefits, athletes’ knowledge of them is sometimes limited [23,24]. In the same way, it has been shown that the use of some SS with less scientific evidence is greater than those with higher levels of supporting research [25]. Finally, some of the main motivators for their consumption are unqualified individuals, such as friends, teammates or the runners themselves [26,27,28,29].

To our knowledge, few studies have been conducted to analyze supplementation patterns in athletes, and no one exclusively in middle-distance runners. Thus, the objective of this research is to know the supplementation trends in those athletes with respect to their level and gender. On the other hand, it aims to assess whether the SS taken by middle-distance runners are those with the most scientific evidence, thus reducing the existing gap in the literature [30].

## 2. Materials and Methods

### 2.1. Type of Study

The research was a descriptive and cross-sectional study. The sample was selected using non-probabilistic, non-injurious and convenience sampling among training groups and individual middle-distance athletes at the national level.

### 2.2. Participants and Study Sample

A total of 106 middle-distance runners (800–1500 m) participated, of which 74 were men and 32 were women (gender assigned at birth). Only two requirements were established to participate in the study, which were as follows: (1) be over 18 years of age (legal age in Spain); (2) be currently performing middle-distance disciplines. The level of the athletes was differentiated by their area of competition, which could be regional (competing at regional or provincial level), national (competitions in Spain) or international (competitions at European and World level). Table 1 describes the age, basic anthropometric data and best performances in middle-distance events of the participants involved in the research.

### 2.3. Instruments

The questionnaire chosen for this research has been previously used in studies with the same objectives carried out in other sports [26,31,32]. This one was chosen for two main reasons; on the one hand, for its contents, structure, applicability and ease of completion for the athletes. The second reason was the quality of the questionnaire, which was created by 25 experts from different areas and achieved a 54% methodological validity, being one of the 57 questionnaires (out of 167) validated to obtain accurate data on supplement consumption [33]. The questionnaire has 4 main parts and a total of 32 questions. The first one asks for personal (e.g., sex), anthropometric (e.g., height, weight) and sociodemographic (e.g., region of residence) data, with a total of 8 questions. The second, with a total of 5 questions, covers topics about the sport practice (e.g., years of practice, level of competition). The third part, with a similar objective, collects information about your best times in the different middle-distance disciplines or about your training days and number of competitions and has a total of 8 questions. Finally, the fourth part (11 questions) covers the area of supplementation, with questions such as: what supplements do they consume, reason for consumption, and place of purchase. This questionnaire collects data about all types of supplements, among which we find sports foods (e.g., energy bars, sports gels), medical supplements (e.g., iron, vitamin D, multivitamins) or performance supplements (e.g., caffeine, creatine, ß-Alanine). These different types of supplements are defined as sport supplements in the current study. From this last section, different questions related to diet were eliminated from the original questionnaire because they did not contribute to the objective of the study and in order to limit the response time. This questionnaire can be obtained in: Suplementación nutricional en la actividad físico-deportiva: análisis de la calidad del suplemento proteico consumido [34].

### 2.4. Procedures

For the data collection, the questionnaire was distributed via training groups, known athletes and social networks. The questionnaire was distributed online so that runners could complete it remotely, voluntarily and anonymously. The protocol complied with the provisions of the Declaration of Helsinki for human research and was approved by the ethical committee of the University of Deusto (ETK-14/23-24) dated 26 October 2023.

### 2.5. Statistical Analysis

To verify whether the variables had a normal distribution, a Kolmogorov–Smirnov test was applied, and Levene’s test was used to verify homoscedasticity. The quantitative data were presented as mean + SD, while the qualitative variables were expressed as percentages and frequencies. A two-way ANOVA was performed for the sex factor (male–female) and level of competition (regional, national and international) to analyze the differences in the total consumption of SS, as well as the SS consumed from the different categories. To assess sex differences, a *t*-test for independent variables was performed, while to assess differences among competition levels, a one-way ANOVA was performed. For those variables in which significant differences were found, the Bonferroni post hoc analysis was used. Regarding the analysis of the athletes who consumed SS, the reason for consumption, the place where they obtained them and who advised them to consume them, a chi-square (χ2) test was used to verify the existence or not of differences between athletes of different sex and level of competition. As for the SS that were consumed by at least 10% of the sample, a χ2 test was performed to verify possible differences according to sex or level of competition. The level of statistical significance was established as *p* < 0.05. The statistical analysis was carried out using the Statistical Package for Social Sciences (SPSS) software v.28.0.0 (IBM, Armonk, NY, USA) for Windows.

## 3. Results

### 3.1. General Consumption of Sport Supplements

Of the total sample, 85.85% reported consuming supplements, while 15 of the 106 subjects responded that they did not consume any type of sport supplement. Regarding sex, supplement consumption was higher in men (89.2%) than in women (78.1%), with no statistical differences between them (*p* = 0.143). In the analysis of the results by level of competition, the percentage of autonomous athletes who consumed supplements was 77.5%, in athletes at the national level it was 90.0%, while in athletes who competed at the international level the consumption was 100%, with no differences between levels (*p* = 0.126).

Table 2 shows the supplements consumed according to the different categories established by the AIS. With respect to total supplement consumption, differences were observed at the competitive level between international and regional athletes (*p* = 0.011). However, no differences were appreciated based on sex (F = 2.248; *p* = 0.466), with a total consumption of 4.8 ± 3.7 and 5.4 ± 5.7 for men and women, respectively. No interactions were observed between level and sex (F = 0.306; *p* = 0.737).

For Group A, no differences were observed between sexes or levels or for the sex–level interaction for the sports food, performance supplement or total intake. However, differences were observed for the group of medical supplements between competition levels (international athletes, *p* = 0.004 vs. regional and *p* = 0.037 vs. national athletes), with consumption being higher as the level of the athletes increased. Likewise, differences between sexes were noted in this group (F = 3.797; *p* = 0.002), with higher consumption in women than in men (0.4 ± 0.6 vs. 0.8 ± 0.9). Table 3 describes the differences between supplement consumption according to level of competition, sex and the interaction between both. Regarding Group B, no differences were observed between sexes (F = 1.591; F = 0.860), levels of competition (F = 2.656; *p* = 0.075) or the interaction between sex and level of competition (F = 0.279; *p* = 0.860). Finally, for group C supplement consumption, differences were observed with respect to sex (F = 13.297; *p* = 0.029), with 0.5 ± 0.6 vs. 0.3 ± 0.4 for males and females, respectively. However, no differences were seen between levels or for the sex–level-of-competition interaction.

### 3.2. Most-Consumed Supplements by Competitive Level and Sex

Table 4 shows those supplements that were consumed by more than 10% of the sample. The most-consumed supplements were caffeine (37%), followed by energy bars and sport drinks (34% for both) and creatine (31.1%). With respect to sex, differences were only observed for iron consumption (*p* < 0.001), with higher consumption in women than in men (17.6% vs. 56.3%). Differences between levels were observed for recovery shakes (83.3% vs. 20% vs. 7.5%, *p* < 0.001; for international, national and regional athletes) and vitamin D (50.0% vs. 18.3% vs. 10.0%, *p* = 0.047; for international, national and regional athletes). The most-consumed supplements in the sport food subgroup for women were sport drinks (34%), contrary to men where the use of sports bars was a little bit higher (36.5%). Regarding medical supplements, iron was the main supplement for both sexes (17.6 vs. 56.3 for male and female).

For performance supplements, differences were observed with caffeine and creatine being the most consumed for men (36.5%) and only caffeine for women (40.6%). Finally, for group C, both BCAA and glutamine were the most-consumed ones for males (12.2%), but not for females (glutamine = 9.4%). As for the level of the athlete, the most-consumed supplements for international athletes were recovery shakes (83.3%), followed by iron and caffeine (66.7%). The national-level athletes’ most-consumed supplements were creatine, sports bars and sport drinks (38.3%), while caffeine was the most-consumed one by regional athletes (35%).

### 3.3. Information about the Place of Purchase, Recommendations and Consumption Patterns

Most athletes took supplements on training and competition days (39.62%). The daily consumption of supplements was 26.42%, followed by training (14.15%) and competition (11.32%). No differences were observed between genders (*p* = 0.106) as opposed to between categories for daily consumption (*p* = <0.00). Thus, 33.3% of the international athletes consumed it daily, while only 18.3% or 7.5% did so in the case of national and regional ones. In analyzing the moment of consumption, most of the sample used them after (56.60%) or before (50.94%) practicing exercise, followed by during training (20.75%). Only a small percentage responded that it was taken during the holiday period (1.89%) or indifferently (7.55%). No differences were observed for levels but between levels for pre- and post-training consumption (*p* = 0.007), which varied according to the level of competition (33% vs. 20% vs. 25% for international, national and regional athletes, respectively).

The principal objective of consumption was to improve performance (70.75%), followed by taking care of their health (35.85%) and palliating dietary deficits (16.98%). Finally, of the 106 middle-distance runners, only 6.60% consumed them for health problems or necessity (3.77%). In this area, no differences were observed between sexes (*p* = 0.564) or levels of competition (*p* = 0.086). The primary place of purchase was the internet (51.89%), followed by specialized stores (26.42%) or a pharmacy (24.54%). Other minority sources of purchase were herbalists (12.26%), sports monitors (3.77%), friends (1.89%) or parapharmacies (0.94%), with no statistically significant differences (*p* = 0.082 and *p* = 0.545 for gender and level). Finally, those who encouraged the use of SS were mainly coaches (37.74%), followed by dieticians–nutritionists (26.42%), teammates (21.70%) or physicians (16.04%). There were other people and sources that recommended its use such as friends and the internet (8.49%) or social network profiles (4.72%). Likewise, there were no differences between levels (*p* = 0.919) or genders (*p* = 0.410).

## 4. Discussion

The main objective of this study was to analyze the supplementation patterns in middle-distance runners, as well as the differences between genders and level of competition. The results indicate that the main differences between levels are observed both in total consumption and in the intake of medical supplements, with these being greater as the level of the athlete increases. Similarly, differences between levels were also observed in the consumption of medical supplements, as well as in pre- and post-training intake. This indicates that, although most athletes place emphasis on performance enhancement via supplementation, higher-level athletes also use these aids to maintain a better state of health and recover between sessions.

Of the total sample, 85.85% responded that they consumed SS, which was higher than the consumption in other disciplines such as fencing or sailing [26,35], but lower than in sports such as rowing, trail running or tennis (100%, 93.8% and 88.6%, respectively) [25,29,31]. No differences were noted for sexes or competition levels, in line with recent research [25,35]. Comparing the data obtained with a sample of athletes from different disciplines, the consumption of SS in middle-distance runners is higher (85% vs. 77%) [36]. Although there have been previous attempts to investigate supplementation patterns in athletes [22], one contained a limited sample while the other had only a few supplements [33,37] and no one has conducted it exclusively in middle-distance athletes. Therefore, this is the first to do so using a representative sample of middle-distance event participants and a broad list of SS.

With respect to total SS consumption, there were differences between international and regional athletes, which had been previously noted in all types of sportsmen and women [33]. With respect to the different groups established by the AIS according to the level of evidence [13], in group A, no differences were observed between levels and genders, contrary to other recent studies [32,35,38]. However, differences between levels were close to being statistically significant (*p* = 0.051). Within the subgroups that exist in group A, only differences in medical supplements are observed for gender, which may be primarily due to the higher consumption of iron among women compared to men (56.3% vs. 17.6%). For the level of competition, differences were also observed in this subgroup, being higher as the level increased. These two findings are contrary to the results from other sports, where no differences have been observed for this subgroup between athlete levels or genders [26,29,35,39].

Regarding group B, which includes SS with emerging evidence but in need of future research, no differences are observed for level and gender, in line with the results in other sports [25,26,29,35,39]. Finally, in group C (supplements with insufficient scientific evidence to support its use), differences were noted between sexes, in line with some [25], but not all, recent evidence [26,35,39]. This could be due to the athlete’s knowledge, which is worse as the level of competition decreases [40].

Taking into account the days of sport practice when they usually take the SS, 39.62% responded that they take them during training and competition, followed by daily consumption and solely on training days, at 26.42% and 14.15%, respectively. Although the main sporting day is similar to that of other sports such as mountain running or rowing [25,29,39], the second and third causes differ between sports. This could be due to differences in the physiological demands of each event, as well as the average duration and energetic requirements of training sessions. Differences were noted in daily consumption between levels of competition, indicating that the main difference between higher- and lower-level athletes was the use of medical supplements on a daily basis. On the other hand, the majority of middle-distance athletes take SS after (56.60%) or before sports practice (50.94%), while a lower percentage take them during sports practice. The duration of middle-distance sessions rarely surpasses 90–120 min [7], while other sports training sessions usually exceed this time, in which they will need to provide higher nutrition and hydration [25]. Here too, differences between levels are observed for pre- and post-consumption, demonstrating how top-level runners place greater importance on preparing for training or recovering for an upcoming workout. In analyzing the reasons for its consumption, the main one is to improve their performance (70.75%), followed by health care (35.85%) and to palliate a dietary deficit (16.98%), similar to other sport disciplines [26,29,31,32,35,36].

Concerning the person who motivated the consumption of SS, the main motivator was the coach (37.74%), followed by dietitians–nutritionists (26.42%), which showed a worse advisor in the case of middle-distance runners compared to other sports [25,39]. The next advisors were teammates, followed by physicians, indicating the existence of other sports modalities with a worse source of support [26,35]. In this sense, dietitians–nutritionists are the most appropriate when choosing one supplement or another regardless of the level of scientific evidence [26,41,42]. Finally, most athletes purchased SS on the internet (51.89%), followed by specialized stores (26.42%) and pharmacies (24.53%). In this sense, both pharmacy and internet products may contain quantities different from those advertised or contaminated substances, which may also put the athletes at risk of unintended doping [43], so athletes tend to go to specialized stores in order to avoid these problems [12,42].

Finally, with regard to the most-consumed SS, we can appreciate caffeine in the first place. Caffeine is a natural stimulant for the central nervous system, possesses various suggested benefits for enhancing performance and is one of the supplements with the highest scientific evidence supporting its use [15]. These advantages encompass enhanced neuromuscular functionality and a decrease in fatigue and perceived effort levels during physical exertion, among others [44]. The following most-consumed SS were sport drinks (formulated to provide a balanced combination of carbohydrates and liquids, facilitating athletes in rehydrating and replenishing energy simultaneously during and after their workout) and sport bars (created as a portable source of carbohydrates, helping meeting carbohydrate intake goals) [13], which also belong to Group A, such as caffeine. These two supplements help mainly in carbohydrate replenishment post or during training or to reach the recommended daily intake of carbohydrates, which can be up to 70% of the total diet or around 6–12 CHO · kg^−1^ · BW · day^−1^. In this sense, carbohydrate intake both during [45] and immediately after [46] exercise limits fatigue and improves performance in the following training sessions.

The next most-consumed supplement was isolated protein, with considerable scientific evidence supporting its use [13], which appears necessary both for the recovery and repair of damaged myofibrillar proteins and to optimize mitochondrial and possibly sarcoplasmic protein synthesis [47]. However, this seems unnecessary in most cases, since athletes tend to consume more protein than any high recommendation [47]. Continuing with the SS that can provide more benefits among those consumed by more than 10% of the sample, we find iron or β-Alanine. Iron plays a fundamental role in the transport of oxygen and a high prevalence of anemia has been observed among middle-distance runners [22]. A small decrease in hemoglobin content (subclinical anemia) leads to a significant decrease in oxygen transport capacity and, therefore, a decrease in performance [48]. Thus, it is important to monitor these variables on a recurring basis in order to supplement if necessary. On the other hand, β-Alanine acts as an intracellular buffer by increasing the concentration of muscular carnosine [49]. Since high-intensity exercise (usually performed by middle-distance runners both in training and competition [7]) increases the amount of hydrogen ions and lowers the intracellular pH from 7.0 to 6.6, supplementation with β-Alanine may improve the ability to withstand this drop, limiting muscular fatigue. However, the determination of whether supplementation enhances performance in elite middle-distance athletes is challenging due to insufficient data and non-performance-related tests [47]. Despite this, considering the absence of side effects and potential performance benefits, individual athletes and their support teams may want to try β-Alanine supplementation to assess its effectiveness for them [1]. Finally, it is important to note the very low percentage of athletes using inorganic nitrates or beetroot juice as SS (6.60%). This supplementation seems to improve performance via the bioavailability of nitric oxide, improving exercise efficiency (decreased O^2^ cost at the same absolute workload) [50]. However, this low use may be due to variability in the response to its supplementation [1] or decreased effects as the physiological capabilities of the athletes increase [50].

It is important to mention that, although a large part of the SS consumed by middle-distance runners in this study belong to group A, it is also observed that there is still a fairly large consumption of supplements with little or no scientific evidence (groups B and C). This has also been observed in other sports, so it is important that athletes use reliable sources of information when deciding which supplements to consume [25,51]. In addition, the present research has several limitations. First of all, the sample is larger than that of other studies with the same population, but a greater participation of international athletes is necessary. In addition, it was the athletes themselves who responded retrospectively to the consumption of SS, which could lead to errors in the number or type of supplements. Therefore, it is necessary to compare and have the support of different federations or institutions worldwide to check if the consumption is similar depending on the competitive level or gender.

## 5. Conclusions

Supplement consumption in middle-distance running is similar to that in other sports. The main differences between levels are seen in the total supplement consumption and in the consumption of medical supplements, as well as in daily or pre- and post-exercise consumption, with these being higher as the level of competition increases. On the other hand, the differences between sexes are found in the consumption of both medical supplements and supplements with limited evidence. Middle-distance runners should improve both their sources of information and places of purchase in order to avoid supplements with low scientific evidence or contaminated/fraudulent products.

## Figures and Tables

**Table 1 nutrients-15-04839-t001:** Characteristics and personal times of the different subjects.

Sex (n)	Category (n)	Age	Height *	Weight *	BMI *	PB 800 m	PB 1500 m
Male (74)	Regional (29)	22.9 ± 5.7	176.3 ± 8.2	64.8 ± 9.1	18.3 ± 2.0	2:01.74 ± 6.80	4:17.22 ± 16.31
	National (43)	24.5 ± 7.6	177.6 ± 6.4	65.1 ± 6.1	18.3 ± 1.4	1:56.42 ± 5.25	4:02.92 ± 16.50
	International (2)	20.0 ± 2.8	189.0 ± 5.7	68.5 ± 4.9	18.1 ± 0.8	1:48.38 ± 1.15	3:47.50
Female (32)	Regional (11)	23.6 ± 8.2	165.0 ± 4.8	52.4 ± 7.0	15.8 ± 1.7	2:26.04 ± 7.04	5:13.66 ± 21.95
	National (17)	21.8 ± 3.1	164.8 ± 4.3	52.5 ± 4.1	15.9 ± 1.1	2:15.56 ± 6.55	4:59.61 ± 43.06
	International (4)	21.0 ± 2.9	167.5 ± 4.8	55.0 ± 3.2	16.4 ± 0.8	2:05.27 ± 3.84	4:15.23 ± 9.97

Results are expressed as mean ± SD. BMI: body mass index. * Self-reported height and weight. BMI calculated from self-reported height and weight. PB: personal best. Gender assigned at birth.

**Table 2 nutrients-15-04839-t002:** Descriptive data of the SS consumed according to the different categories defined by the AIS as a function of sex and level of competition.

Variable		Sex		Level of Competition							
		M	F	R	N	I			Total		
		Mean ± SD	Mean ± SD	Mean ± SD	Mean ± SD	Mean ± SD	Med	IQ	Mean ± SD	Med	IQ
	Total SS	4.8 ± 3.7	5.4 ± 5.7	4.0 ± 3.8	5.1 ± 3.7	9.5 ± 9.6	6.0	26	5.0 ± 4.4	5.0	27
Group A	Sports food	1.1 ± 1.0	1.0 ± 1.2	1.0 ± 1.1	1.1 ± 1.0	1.2 ± 1.5	0.5	3	1.1 ± 1.1	1.0	4
	Medical supplement	0.4 ± 0.6	0.8 ± 0.9	0.3 ± 0.5	0.6 ± 0.8	1.3 ± 1.0	1.0	3	0.52 ± 0.7	0.0	3
	Performance supplement	1.0 ± 1.1	0.9 ± 1.1	0.8 ± 1.1	1.0 ± 1.9	1.7 ± 1.6	1.5	4	1.0 ± 1.1	1.0	4
	Total Group A	2.5 ± 1.9	2.7 ± 2.4	2.1 ± 2.0	2.7 ± 1.9	4.2 ± 3.7	3.0	10	2.5 ± 2.1	2.0	10
	Group B	0.5 ± 0.7	0.5 ± 0.7	0.5 ± 0.6	0.5 ± 0.7	1.2 ± 1.2	1.0	3	0.5 ± 0.7	0.0	3
	Group C	0.5 ± 0.6	0.3 ± 0.4	0.3 ± 0.6	0.5 ± 0.6	0.4 ± 0.6	0.5	1	0.4 ± 0.6	0.0	2

AIS: Australian Institute of Sport; SS: sport supplements; SD: standard deviation; M: male; F: female; R: regional; N: national; I: international; Group A: supplements with solid scientific evidence in specific situations under established protocols; Group B: components with emerging evidence that should be used in research or clinical settings; Group C: supplements with limited evidence and effects on performance; gender assigned at birth.

**Table 3 nutrients-15-04839-t003:** ANOVA of the SS consumed according to the different categories defined by the AIS as a function of sex, level of competition and their interaction.

Variable		Sex		Level of Competition		Sex–Level-of-Competition (Mean ± SD)							
		F	*p*	F	*p*	R		N		I		F	*p*
						M	F	M	F	M	F		
	Total SS	2.248	0.466	4.582	0.012 ^#^	4.0 ± 3.6	3.9 ± 4.5	5.1 ± 3.4	5.2 ± 4.5	7.5 ± 9.1	10.5 ± 11.1	0.306	0.737
Group A	Sports food	1.330	0.726	0.102	0.903	1.1 ± 1.1	0.8 ± 1.1	1.1 ± 0.9	1.1 ± 1.4	1.5 ± 2.1	1.0 ± 1.4	0.270	0.764
	Medical supplement	3.797	0.002 *	5.693	0.005 ^#$^	0.24 ± 0.4	0.6 ± 0.7	0.5 ± 0.7	0.8 ± 0.9	0.5 ± 0.7	1.8 ± 1.0	1.138	0.325
	Performance supplement	0.014	0.592	2.167	0.120	0.8 ± 1.1	0.6 ± 1.3	1.1 ± 1.1	0.8 ± 0.7	1.5 ± 2.1	1.8 ± 1.7	0.162	0.850
	Total group A	0.671	0.560	3.066	0.051	2.1 ± 2.0	2.0 ± 2.3	2.7 ± 1.8	2.8 ± 2.1	3.5 ± 4.9	4.5 ± 3.7	0.163	0.849
	Group B	1.591	0.860	2.656	0.075	0.5 ± 0.7	0.5 ± 0.5	0.5 ± 0.8	0.4 ± 0.5	1.0 ± 1.4	1.3 ± 1.3	0.279	0.757
	Group C	13.297	0.029 *	1.884	0.157	0.3 ± 0.7	0.1 ± 0.3	0.6 ± 0.6	0.3 ± 0.5	0.5 ± 0.7	0.5 ± 0.6	0.151	0.860

AIS: Australian Institute of Sport; SS: sport supplements; SD: standard deviation; M: male; F: female; R: regional; N: national; I: international; Group A: supplements with solid scientific evidence in specific situations under established protocols; Group B: components with emerging evidence that should be used in research or clinical settings; Group C: supplements with limited evidence and effects on performance; gender assigned at birth. * Statistical difference at *p* < 0.05 between male and female. ^#^ Statistical difference at *p* < 0.05 between regional and international athletes. ^$^ Statistical difference at *p* < 0.05 between national and international athletes.

**Table 4 nutrients-15-04839-t004:** Distribution (%) of the most-consumed supplements (>10%) as a function of sex and level of competition according to the categories defined by the AIS.

Category		Supplement Name	Total (%)	Sex (%)			Level of Competition (%)			
				M	F	*p*	R	N	I	*p*
Group A	Sports foods	Sport bars	34.0	36.5	28.1	0.273	30.0	38.3	16.7	0.451
		Sport drinks	34.0	33.8	34.4	0.561	27.5	38.3	33.3	0.533
		Sports gel	21.7	21.6	21.9	0.582	22.5	21.7	16.7	0.949
		Whey protein	30.2	29.7	31.3	0.525	25.0	31.7	50.0	0.429
		Recovery shakes	18.9	20.3	15.6	0.394	7.5	20.0	83.3	<0.001 *
	Medical supplements	Iron	29.2	17.6	56.3	<0.001 *	22.5	30.0	66.7	0.084
		Vitamin D	17.0	14.9	21.9	0.269	10.0	18.3	50.0	0.047 *
	Performance supplements	β-Alanine	20.8	20.3	21.9	0.521	12.5	23.3	50.0	0.081
		Caffeine	37.7	36.5	40.6	0.424	35.0	36.7	66.7	0.318
		Creatine	31.1	36.5	18.8	0.054	20.0	38.3	33.3	0.151
Group B		Vit C	19.8	20.3	18.8	0.542	17.5	20.0	33.3	0.662
Group C		BCAA	10.4	12.2	6.3	0.295	10.0	10.0	16.7	0.873
		Glutamine	11.3	12.2	9.4	0.482	5.0	15.0	16.7	0.276

AIS: Australian Institute of Sport; M: male; F: female; R: regional; N: national; I: international; Group A: supplements with solid scientific evidence in specific situations under established protocols; Group B: components with emerging evidence that should be used in research or clinical settings; Group C: supplements with limited evidence and effects on performance; gender assigned at birth. * Statistical difference at *p* < 0.05.

## Data Availability

Data of the article are available in the tables of this paper or on request from the corresponding author.

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
