# Peer review of "Are Supplements Consumed by Middle-Distance Runners Evidence-Based? A Comparative Study between Level of Competition and Sex"

_nutrients, 2023, doi:10.3390/nu15224839_

Round 1
Reviewer 1 Report
Comments and Suggestions for Authors
Dear authors,
Congratulations on the paper submitted. My few comments aimed to improve the content of the article. The gap in the literature can be improved by adding more literature on sports nutrition.
[1] L66. The gap in the literature should be defined as most of the studies among athletes focused on nutritional aspects and evaluate the benefits of supplementations regarding important parameters in sport participation (i.e., performance, body composition, injury prevention). Please, consider to cite the following reference:
Fleming JA, Naughton RJ, Harper LD. Investigating the Nutritional and Recovery Habits of Tennis Players. Nutrients. 2018;10(4):443. Published 2018 Apr 3. doi:10.3390/nu10040443
[2] L103. The authors should define all sections of the questionnaire.
[3] What was the scale of the instrument? This should be stated in the methods.
[4] The meaning of letters should be stated as a footnote in Table 2.
[5] The quality of the tables could be improved but the results are well written.
[6] The first paragraph of the discussion should summarize the main results of the study.
Author Response
Dear reviewers,
We are glad that you found the article interesting and that you consider that it can contribute to close the existing gap in the knowledge of sports nutrition and supplementation. We would like to thank you for the contributions and suggestions and then we will answer your specific comments. On the other hand, we underline in yellow the changes or introductions made to make it easier to read.
1st Reviewer:
[1] L66. The gap in the literature should be defined as most of the studies among athletes focused on nutritional aspects and evaluate the benefits of supplementations regarding important parameters in sport participation (i.e., performance, body composition, injury prevention). Please, consider to cite the following reference:
Fleming JA, Naughton RJ, Harper LD. Investigating the Nutritional and Recovery Habits of Tennis Players. Nutrients. 2018;10(4):443. Published 2018 Apr 3. doi:10.3390/nu10040443
This has been added and underlined in yellow in the manuscript.
[2] L103. The authors should define all sections of the questionnaire.
Dear reviewer, all questionnaire sections are described in the text (lines 95 to 105).
[3] What was the scale of the instrument? This should be stated in the methods.
The questionnaire has no scales in the supplements section, it gives the possibility to choose which supplements the athlete has consumed, time of consumption, place of purchase, advisor...
[4] The meaning of letters should be stated as a footnote in Table 2.
Thank you for the comment. The meaning has been added in the notes below the table.
[5] The quality of the tables could be improved but the results are well written.
Dear reviewer. When we first uploaded the documents to the system they all were in the correct format but they have changed when the document was converted to the journal format. Anyway, the tables have been attached as another document to the journal.
[6] The first paragraph of the discussion should summarize the main results of the study.
The main findings have been summarized and added underlined in yellow in the first paragraph of the discussion.
Reviewer 2 Report
Comments and Suggestions for Authors
There are a couple of typos error:
Line 40 and 259: "middle- distance.
Paragraph 2.5 has different layout from the rest of the article.
Table 2 layout MUST be redone: awfully reading.
Table 3 layout must be improved.
Table 4: 4 over 5 of the caption are in capital letters. All the others are in lower.
Comments on the Quality of English Language
Good
Author Response
Dear reviewers,
We are glad that you found the article interesting and that you consider that it can contribute to close the existing gap in the knowledge of sports nutrition and supplementation. We would like to thank you for the contributions and suggestions and then we will answer your specific comments. On the other hand, we underline in yellow the changes or introductions made to make it easier to read.
2nd Reviewer:
Line 40 and 259: "middle- distance.
Those errors have been corrected and underlined in yellow.
Paragraph 2.5 has different layout from the rest of the article.
Dear reviewer, as with the tables, this one was in the correct format when we uploaded all the files into the journal’s system, but was changed when it was transformed to the journal document.
Table 2 layout MUST be redone: awfully reading.
The tables have been attached as another document to the journal. When we sent them, they were in the correct format but they have changed when the document was converted to the journal’s format.
Table 3 layout must be improved.
As previously mentioned, the tables have been attached as another document to the journal. When we sent them, they were in the correct format but they have changed when the document was converted to the journal’s format.
Table 4: 4 over 5 of the captions are in capital letters. All the others are in lower.
Thank you for the comment. These mistakes have been corrected and are underlined in yellow.

Reviewer 3 Report
Comments and Suggestions for Authors
This is an interesting article which provides valuable insights for coaches and other professionals working with athletes. My comments primarily focus on the writing; specifically, the need for a thorough copyedit by an native English speaker. This can help address inconsistencies in the paper, such as the use of initial caps, period vs. comma in percentages, etc.
Specific comments follow:
Line 11: Add running after Middle distance (Middle-distance running events…)
Line 16-17: Recommend rewriting sentence to something like this: In this descriptive, cross-sectional study data was collected from 106 middle-distance runners using a validated questionnaire.
Lines 17-19: The wording of this sentence “Of the total sample, 85.85% responded that they consumed supplements, with no differences for level of competition or sex” says the athletes took the supplements for both competition and sex. I’m think you mean to say the following: Of the total sample, 85.85% responded that they consumed supplements; no statistical difference was found regarding level of competition or sex of the athletes.
Line 30: Same note as line 11.
Line 31: Energetic level sounds awkward. Perhaps something like The level of energy intensity is in a middle ground…
Line 40: Remove extra space after hyphen: middle- distance
Line 58-59: Make c in Caffeine lower case, same for Sodium Bicarbonate. Also delete “the” before sodium bicarbonate. Lastly, what is the level of evidence supporting creatinine or hydrolyzed collagen? If good, suggest adding these two supplements to the list.
Line 60: What supplements are commonly used among this class of athletes?
Line 78: Add in parentheses (gender assigned at birth)—if that is true.
Table 1: Same comment as for line 78. Add to table note if true.
Line 88: Instruments. Add a sentence about the survey collecting data on both sport foods such as [give examples or define], medical supplements [give examples or define], and performance supplements [give examples or define]. Also indicate that for this study, sport supplements is employed as an overarching term for these three classes of supplements. Is that correct?
Line 89: Does the questionnaire have a name? If so, add it. Also, instead of the current references, add the validity study as your source. I think this is it, however, my Spanish language skills are very limited: Sánchez Oliver AJ. Suplementación nutricional en la actividad físico-deportiva: análisis de la calidad del suplemento proteico consumido. Granada (ES): Universidad de Granada; 2013.
Line 103: Should this be a period? place of purchase...
Lines 114-139: Different font.
Line 118: What is SS? Is it sport supplements? If so, define acronym at first usage.
Line 140: Define AIS acronym.
Table 2: Define AIS. Explain differences between groups in notes.
Table 3: The text for Group A is cut off. Explain differences between groups in notes.
Lines 187-192. Before decimal in percentages consistently use period vs. comma.
Line 216: Change “internet” to the Internet.
Line 270: Make initial caps lower case in Dietitian-Nutritionist (dietitian-nutritionist).
Line 277: Change internet to Internet.
Line 279: Add note, that those products may also put the athletes at risk of unintended doping:
Martínez-Sanz JM, Sospedra I, Ortiz CM, Baladía E, Gil-Izquierdo A, Ortiz-Moncada R. Intended or Unintended Doping? A Review of the Presence of Doping Substances in Dietary Supplements Used in Sports. Nutrients. 2017; 9(10):1093. https://doi.org/10.3390/nu9101093
Line 289: Make c in caffeine lower case.
Comments on the Quality of English Language
My comments primarily focus on the writing; specifically, the need for a thorough copyedit by an native English speaker. This can help address inconsistencies in the paper, such as the use of initial caps, period vs. comma in percentages, etc.
Author Response
Dear reviewers,
We are glad that you found the article interesting and that you consider that it can contribute to close the existing gap in the knowledge of sports nutrition and supplementation. We would like to thank you for the contributions and suggestions and then we will answer your specific comments. On the other hand, we underline in yellow the changes or introductions made to make it easier to read
3rd Reviewer:
Line 11: Add running after Middle distance (Middle-distance running events…)
Thank you for the comment. It has been added and underlined in yellow in the text.
Line 16-17: Recommend rewriting sentence to something like this: In this descriptive, cross-sectional study data was collected from 106 middle-distance runners using a validated questionnaire.
Amended. The sentence has been replaced by the proposed option.
Lines 17-19: The wording of this sentence “Of the total sample, 85.85% responded that they consumed supplements, with no differences for level of competition or sex” says the athletes took the supplements for both competition and sex. I’m think you mean to say the following: Of the total sample, 85.85% responded that they consumed supplements; no statistical difference was found regarding level of competition or sex of the athletes.
The proposed sentence has been changed by the reviewer's recommendation.
Line 30: Same note as line 11.
Amended. It has been added and underlined in yellow.
Line 31: Energetic level sounds awkward. Perhaps something like The level of energy intensity is in a middle ground…
The proposed sentence has been changed per the reviewer's recommendation
Line 40: Remove extra space after hyphen: middle- distance
Amended. It has been corrected and underlined in yellow.
Line 58-59: Make c in Caffeine lower case, same for Sodium Bicarbonate. Also delete “the” before sodium bicarbonate. Lastly, what is the level of evidence supporting creatinine or hydrolyzed collagen? If good, suggest adding these two supplements to the list.
Thank you. These errors have been corrected and underlined in yellow in the text. After searching the literature, we found that there is not much evidence to support the use of both creatinine and hydrolyzed collagen in sport. Also, in the questionnaire there is a question about whether they take any supplements that are not on the supplement list and none of the participants answered that they consumed creatinine or hydrolyzed collagen.
Line 60: What supplements are commonly used among this class of athletes?
A sentence has been added “with the consumption of vitamin, minerals and aminoacids being higher than the previous mentioned ones”, as the research indicates.
Line 78: Add in parentheses (gender assigned at birth)—if that is true.
It has been added and underlined in yellow in the text.
Table 1: Same comment as for line 78. Add to table note if true.
As in line 78, it has been added in all the tables and underlined in yellow.
Line 88: Instruments. Add a sentence about the survey collecting data on both sport foods such as [give examples or define], medical supplements [give examples or define], and performance supplements [give examples or define]. Also indicate that for this study, sport supplements is employed as an overarching term for these three classes of supplements. Is that correct?
Yes, for the study sport supplements is used for the three terms. This fact, has been added and underlined in yellow.
Line 89: Does the questionnaire have a name? If so, add it. Also, instead of the current references, add the validity study as your source. I think this is it, however, my Spanish language skills are very limited: Sánchez Oliver AJ. Suplementación nutricional en la actividad físico-deportiva: análisis de la calidad del suplemento proteico consumido. Granada (ES): Universidad de Granada; 2013.
Thank you for the comment. It has been added as a reference.
Line 103: Should this be a period? place of purchase...
Dear reviewer, with the term place of purchase we wanted to know the place where the athletes obtained the supplementation (pharmacies, internet, specialised shops, etc.). Each athlete was responsible for marking the option(s) they used.
Lines 114-139: Different font.
Yes, you are right. As with the tables, this was in correct format when we uploaded the files to the journal and in the conversion to the format some problem has occurred, we are sorry for the inconvenience. The editor is aware of this fact.
Line 118: What is SS? Is it sport supplements? If so, define acronym at first usage.
This acronym has been defined in the abstract and used throughout the text (including tables), in addition, it has been included in the list of abbreviations.
Line 140: Define AIS acronym.
This acronym has also been defined in the introduction and used throughout the text (including tables), in addition, it has been included in the list of abbreviations.
Table 2: Define AIS. Explain differences between groups in notes.
This acronym has been defined in notes.
Table 3: The text for Group A is cut off. Explain differences between groups in notes.
The tables are attached in another document to the journal. When we sent it, they were in the correct format but they have changed when the document was converted to the journal format. The differences have been explained in notes also.
Lines 187-192. Before decimal in percentages consistently use period vs. comma.
Amended. Commas have been replaced by periods.
Line 216: Change “internet” to the Internet.
Amended. The first letter has turned into a capital letter.
Line 270: Make initial caps lower case in Dietitian-Nutritionist (dietitian-nutritionist).
Amended. The initial caps have been has turned into a lower case letters.
Line 277: Change internet to Internet
The first letter has turned into a capital letter.
Line 279: Add note, that those products may also put the athletes at risk of unintended doping:
Martínez-Sanz JM, Sospedra I, Ortiz CM, Baladía E, Gil-Izquierdo A, Ortiz-Moncada R. Intended or Unintended Doping? A Review of the Presence of Doping Substances in Dietary Supplements Used in Sports. Nutrients. 2017; 9(10):1093. https://doi.org/10.3390/nu9101093
This reference has been added and underlined in yellow.
Line 289: Make c in caffeine lower case.
Amended. The first letter has turned into a capital letter.
